# Capsaicin 8% Patch and Chronic Postsurgical Neuropathic Pain

**DOI:** 10.3390/jpm11100960

**Published:** 2021-09-27

**Authors:** Luca Gregorio Giaccari, Caterina Aurilio, Francesco Coppolino, Maria Caterina Pace, Maria Beatrice Passsavanti, Vincenzo Pota, Pasquale Sansone

**Affiliations:** Department of Woman, Child and General and Specialized Surgery, University of Campania “Luigi Vanvitelli”, 80138 Napoli, Italy; lucagregorio.giaccari@gmail.com (L.G.G.); Caterina.aurilio@unicampania.it (C.A.); francesco.coppolino1987@gmail.com (F.C.); caterina.pace@libero.it (M.C.P.); beatrice.passavanti@libero.it (M.B.P.); vincenzo.pota@inwind.it (V.P.)

**Keywords:** chronic post-surgical pain (CPSP), capsaicin 8% patch, Qutenza^®^

## Abstract

(1) **Background**: Surgery is a frequent cause of persistent pain, defined chronic post-surgical pain (CPSP). The capsaicin 8% patch (Qutenza^®^) is approved for the treatment of postherpetic neuralgia (PHN) and for diabetic peripheral neuropathy (DPN) of the feet. We propose a review of the literature on use of the capsaicin 8% patch to treat neuropathic pain associated with surgery; (2) **Methods**: We identified the articles by searching electronic databases using a combination of such terms as “*capsaicin 8% patch*”, “*Qutenza^®^*”, and “*c**hronic postsurgical pain*”; (3) **Results**: We identified 14 selected studies reporting on a total of 632 CPSP cases treated with capsaicin 8% patch. Treatment with the capsaicin 8% patch significantly reduced the average pain intensity. Only 5 studies reported adverse events (AEs) after the patch application. The most common AEs were erythema, burning sensation and pain; (4) **Conclusions**: Our review indicate that capsaicin 8% patch treatment for CPSP is effective, safe and well tolerated, but randomized controlled trials on efficacy, safety and tolerability should be conducted.

## 1. Introduction

Surgery is a frequent cause of persistent pain [1]. Chronic post-surgical pain (CPSP) was first defined by Macrae and Davies in 1999 [2], and then expanded by Macrae in 2001 [3], as “*pain that develops after surgical intervention and lasts at least 2 months”.* In 2014, Werner and Kongsgaard proposed an updated definition [4]. According to the authors, “*the pain develops at least 3–6 months after a surgical procedure or increases in intensity after the surgical procedure. The pain is either a continuation of acute post-surgery pain or develops after an asymptomatic period. It is either localized to the surgical field or referred to a dermatome”.* In all cases, other causes of the pain should be excluded.

According to data on CPSP from the European registry PAIN OUT (Improvement in postoperative PAIN OUTcome), the incidence of moderate to severe CPSP at 12 months is 11.8% [5]. The incidence of CPSP varies according to the surgical procedure performed: specifically after open cholecystectomy and total knee arthroplasty the incidence of severe CPSP is high [5]. Various risk factors have been attributed to CPSP: the type and approach of surgery, expecially after thoracotomy, cardiac surgery, hernia surgery, mastectomy and amputations; patient factors including female gender and being a young adult; preexisting patient conditions such as pain present preoperatively and any preexisting painful conditions in other parts of the body; perioperative factors including duration and type of surgery, extent of nerve damage intraoperatively, and severity and duration of acute postoperative pain [5,6].

There is a lack of evidence for the treatment of CPSP [7]. The evidence for treatment options for established CPSP is extremely limited. High-quality trials of multimodal interventions matched to pain characteristics are needed to provide robust evidence to guide management of CPSP [8]. As first-line treatment, the Neuropathic Pain Special Interest Group (NeuPSIG) from the International Association for the Study of Pain (IASP) has recommended tricyclic antidepressants (TCA) and serotonin-norepinephrine reuptake inhibitors (SNRI). Lidocaine patches, capsaicin patches and tramadol are the second-line treatment, while opioids (particularly oxycodone and morphine) and botulinum toxin are the ultimate therapeutic option [9].

Capsaicin is a highly selective agonist of transient receptor potential vanilloid type 1 (TRPV1). They are found in the nerves in the skin that detect pain and they are responsible for pain hypersensitivity in neuropathic pain [10]. High doses or prolonged exposure to capsaicin induce the defunctionalization of TRPV1. This desensitization involves mechanisms not entirely understood, including depletion of neuropeptides such as substance P in the nerve fibers that express TRPV1 and an increase of intracellular calcium levels.

The capsaicin 8% patch (Qutenza^®^) contains a total of 179 mg of capsaicin (640 mcg per cm^2^) [11]. Qutenza^®^ is approved in the EU for the treatment of peripheral neuropathic pain (PNP) in adults [12]. It is indicated in adults for the treatment of postherpetic neuralgia (PHN) and for diabetic peripheral neuropathy (DPN) of the feet [11,13]. The rapid release of high doses of capsaicin overstimulates the TRPV1, so that it becomes less sensitive to the stimuli that normally cause neuropathic pain. Several factors influence the treatment: female patients with low baseline pain intensity with the presence of hypoesthesia and no allodynia are more likely to achieve pain relief. Treatment with the capsaicin 8% patch must be performed only by a healthcare provider [11]. Qutenza is not for use near eyes or mucous membranes and it can cause serious side effects, including local pain, redness, itching or increases in blood pressure during or right after treatment [11].

**Aims.** To our knowledge, limited or no evidence based on clinical trials is available on the efficacy of the capsaicin 8% patch for peripheral neuropathic pain different from post-herpetic neuralgia and HIV-associated neuropathy in non-diabetic patients or in diabetic polyneuropathy [14,15].

In this article, we propose a review of the literature on use of the capsaicin 8% patch to treat neuropathic pain associated with surgery. The aim is to review the evidence from studies on the efficacy and tolerability of topically applied high-concentration capsaicin for CPSP in adults.

## 2. Materials and Methods

**Protocol and registration.** We performed a systematic review based on the Preferred Reporting Items for Systematic Reviews and Meta-Analyses (PRISMA) statement [16]. The protocol was not published but it is available upon request. The review was not registered with the International prospective register of systematic reviews (PROSPERO).

**Literature search.** We identified the articles by searching electronic databases (Embase, MEDLINE, Google Scholar and Cochrane Central Trials Register). Other relevant studies were identified from the reference lists. We used a combination of such terms as “***capsaicin 8% patch***”, “*Qutenza^®^*”, and “*chronic postsurgical pain*”.

The initial search was performed on 1 June 2021.

The titles and abstracts were screened by two researchers (LGG and PS) to identify the key words. The selected papers were read in full by the same two independent reviewers (LGG and PS) and a third reviewer (MCP) was consulted in case of disagreement.


**Inclusion and Exclusion Criteria.**


Studies were included if they met all of the following criteria:—the full study was published;—the study described clinical use of 8% capsaicin patch for CPSP;—the study reported the clinical outcome of the patient(s) treated with 8% capsaicin patch.

Reasons for exclusions were:

—studies published only as short abstracts (usually meeting reports) or studies of experimentally induced pain;—the study did not report clinical outcome;—the study had duplicate data with others (in these cases, only the largest study was retained);—the study presented pooled data that did not allow for extrapolation of useful information.

**Data extraction and management.** Data were extracted independently by one of the three reviewers (LGG, PS, MCP) according to a predefined protocol. The data extraction was then checked by one of the other two reviewers, and discrepancies were solved by discussion among all of them.

Variables of interest included:

—demographic characteristics (sex and age);—clinical characteristics;—CPSP aetiology;—therapeutic regimen and dosage;—adverse events (AEs);—therapeutic outcome.

Clinical characteristics were reported together with the ratio of the number of patients in whom the variable was present (n) and the total number of reported cases (N): n/N (%). As for the symptoms, we assumed they were absent rather than missing if they were not cited in the manuscript, in order to account for the reporting bias, and therefore described as zero (n) out of the total number of reported cases (N).

## 3. Results

**Study selection.** Our research identified 101 studies in PUBMED, 326 in Google Scholar and 3 in Cochrane Library. After the duplicate studies were removed, we identified 41 articles in the researched databases, and 14 of them were included in our systematic review. The 14 selected studies reported on a total of 632 CPSP cases treated with capsaicin 8% patch: 8 of them were observational studies (601 patients), 5 case report/case series (7 patients) and 1 randomized control trial (24 patients).

Participants had pain due to CPSP.

The flow diagram (see Figure 1) shows the results from the literature search and the study selection process.

**Study characteristics.** In Table 1, all the studies are presented alphabetically with a brief clinical description per case.

All the cases were from European countries, except for Roberts C et al.: 4 cases were from Germany [18,19,23], 2 from Portugal [22,26], 1 from Belgium [17], Spain [21], France [29], Greece [30], Ireland [25], and Italy [28]. Mankowski C et al. conducted the ASCEND study involving seven European countries.

**Patient Characteristics.** The median age of the study populations varied between 24 and 65 years. The majority of patients were female (61.9%) and the male:female ratio of all studies combined was 0.62. For 3 studies was not possible to report the patients’ characteristics [17,21,23].

The population is further described in Table 2.

CPSP of various aetiology (thoracotomy or thoracoscopy, 15.1% [*n* = 56]; knee surgery, 12.9% [*n* = 48]; parietal surgery, 11.3% [*n* = 42]; mastectomy, 4.0% [*n* = 15]; back surgery, 1.6% [*n* = 6]; traumatic surgery, 1.3% [*n* = 5]; abdominal surgery, 1.1% [*n* = 4]; head surgery, 0.5% [*n* = 2]; and not specified, 52.2% [*n* = 194]) occurred in all cases.

CPSP aetiology is reported in Table 3.

Patients received a wide range of previous analgesic medications (see Table 4): acetaminophen (3 studies [18,27,28]), anticonvulsivants, such as gabapentin and pregabalin (8 studies [18,19,20,21,22,27,28,30]), antidepressants (7 studies [18,19,20,21,22,27,30]), interventional techniques, such as neuroaxial blocks, intercostal blocks or sympathetic blocks (1 study [21]), NSAIDs (2 studies [18,21]), opioids (7 studies [18,19,20,21,22,28,30]), and topical lidocaine (3 studies [20,21,22]).

As shown in Table 2, the following regions were treated with capsaicin 8% patch: head (1.7%, *n* = 7), chest (20.7%, *n* = 85), back (1.7%, *n* = 7), abdomen (12.7%, *n* = 52), upper limbs (1.2%, *n* = 5) and lower limbs (15.1%, *n* = 62). Other or non-specified body regions were treated in 46.8% of patients (*n* = 192). Mankowski C et al. and Pinto TJ et al. did not specified painful body regions.

**Pain intensity.** In all studies, pain was of at least moderate severity and was frequently unresponsive to, or poorly controlled by, conventional therapy. Treatment with the capsaicin 8% cutaneous patch significantly reduced the average pain intensity. The NRS baseline score of 6.9 decreased to all subsequent visits.

The duration of application of capsaicin 8% patch varied between 30 and 65 min, with most patients treated for 60 min. Each patch was applied for 30 min to the feet and for 60 min to other areas of the body, as recommended [11]. Roberts C et al. treated a patient with persistent pain post-thoracotomy for 65 min.

In 7 studies more than one application was required [19,20,24,25,26,29,30].

Baseline pain intensity is reported in Table 5.

**Adverse Events.** Only 5 studies reported adverse events (AEs) after the capsaicin 8% cutaneous patch application [17,18,21,29,30]. Where AEs were not reported, we considered them absent. The exact number of patients who experienced adverse events could not be determined as more than one AE may appear in a single individual.

The most common AEs were application site reactions. The main AEs were erythema (*n* = 91), burning sensation (*n* = 78) and pain (*n* = 35).

A local anaesthetic pre-treatment was reported in 8 studies [17,18,19,20,21,23,27,29]. The most frequently used local anaesthetic was 4% lidocaine, or 2.5% lidocaine in combination with 2.5% prilocaine (EMLA^®^). Subsequent cooling was carried out in 4 studies [17,18,19,21], and patients were told to use rescue medications after patch application. In the other studies, no pre-treatment was done.

No systemic AEs were reported.

Treatment discontinuation was reported by Bischoff et al. A patient experienced severe pain at the application site, which necessitated patch removal, and was withdrawn from the study.

## 4. Discussion

Between 20 and 56% of patients undergoing surgery develop chronic pain. Patients may develop CPSP after “common surgical procedures”. The majority of included patients in our review were post-thoracotomy (15.1%). Thoracotomy is reported as one of the most common causes of CPSP with an incidence of 20–40% [31]. Other common surgeries include limb amputation [32], mastectomy [33], and inguinal hernia repair [34].

Tricyclic antidepressants and anticonvulsivants are recommended as first-line therapy, but systemic adverse effects may limit their use particularly in elderly patients [35]. In 2015 the NeuPSIG recommendations concluded that high-concentration capsaicin patches can be as second-line therapy used for localized neuropathic pain syndromes where a presumed local pain generator existed or where there are concerns regarding tolerability of oral therapy [9].

Capsaicin 8% patch is indicated in adults for the treatment of neuropathic pain associated with postherpetic neuralgia (PHN) and for neuropathic pain associated with diabetic peripheral neuropathy (DPN) of the feet [11]. Most of the findings on the efficacy of high-dose capsaicin in chronic neuropathic pain syndromes have been collected in patients with post-herpetic neuralgia [14,15], HIV polyneuropathy [14,15], and diabetic neuropathy [14,15].

The efficacy of 8% capsaicin patch for the treatment of peripheral neuropathic pain has been investigated in several trials in patients with CPSP. The efficacy of single and multiple applications of the capsaicin 8% patch in patients with CPSP was assessed in this review. Off label use of the capsaicin 8% patch in relieving CPSP proved temporarily effective.

It should be noted that the capsaicin 8% patch has been shown to be also effective in patients with CPSP localized to the face, although current recommendations restrict the application of the patch to this region [11]. Gaul C et al. reported two cases of chronic pain after the removal of a benign tumour of the parotid and the extraction of a wisdom tooth successfully treated with capsaicin 8% patch. In this context, the high-concentration capsaicin patch is a further treatment option, although there are special warnings and precautions for the use in the treatment of head or facial areas.

Only in a study included in this review no significant difference in pain reduction was demonstrated comparing the pain relief of a capsaicin 8% patch with an inactive placebo patch. Bischoff JM et al. considered patients with CPSP due to inguinal herniorrhaphy. The authors suppose that the interruption of cutaneous nerve transmission alone may be insufficient to relieve postherniorrhaphy pain and seem to indicate that nerve injury, as well as deep tissue inflammation, may contribute to the development and maintenance of persistent postherniorrhaphy pain.

In our review, the capsaicin 8% patch was generally well tolerated in patients with CPSP. Of the 632 patients receiving the capsaicin 8% patch in the studies included in this review, 32.3% reported treatment-related AEs. Most AEs were self-limiting and mild or moderate in severity. No AEs led to discontinuation in patients treated with the capsaicin 8% patch. The most frequently reported AEs were transient application-site reactions. Only burning sensations, erythema and local pain were reported as side effects, the treatment of which was easily manageable. Other frequently reported AEs (<5% of patients) such as nausea, nasopharyngitis, bronchitis, sinusitis, vomiting, pruritus or hypertension were not reported [11].

Use of local anaesthetic before application, local cooling, and availability of short-acting opioids for pain relief in the first few days following treatment all help to increase tolerability of the treatment.

Capsaicin has the advantage of site-specific delivery with lower total systemic dose and avoidance of first-pass metabolism, reducing the risk of drug interactions and adverse effects. In fact, capsaicin 8% patch was well tolerated with only minor application site events by our patients.

**Limitations.** Recommendations on the effectiveness of capsaicin patch could not be drawn due to the studies design included in our review. This study was based on observational studies, the majority of which were case series or case reports. Due to the greater potential for bias, they are often excluded from systematic reviews of treatments. We consider case series and observational studies essential to contribute to the available evidence base, and their results supplement the limited evidence available from other studies. We did not perform a meta-analysis due to the design of most of the studies (case report, case series) and the lack of a comparator.

## 5. Conclusions

Persistent postoperative pain affects millions of patients every year; it is a potential burden for the healthcare systems that until recently has been an unrecognized complication of surgery. What we can do to prevent patients from developing chronic pain postsurgery is now considered one of the most important research priorities in anesthesia and perioperative medicine.

To date and to our knowledge, however, only very little data is available on the efficacy of high-dose capsaicin therapy in patients with CPSP, although the treatment of neuropathic pain in these patients is a particular challenge due to the complexity and diversity of symptoms. CPSP is challenging to manage because patients are often non-compliant or unable to tolerate recommended existing systemic treatments. Our review indicate that 8% capsaicin patch treatment for CPSP is effective, safe and well tolerated, but randomized controlled trials on efficacy, safety and tolerability should be conducted.

## Figures and Tables

**Figure 1 jpm-11-00960-f001:**
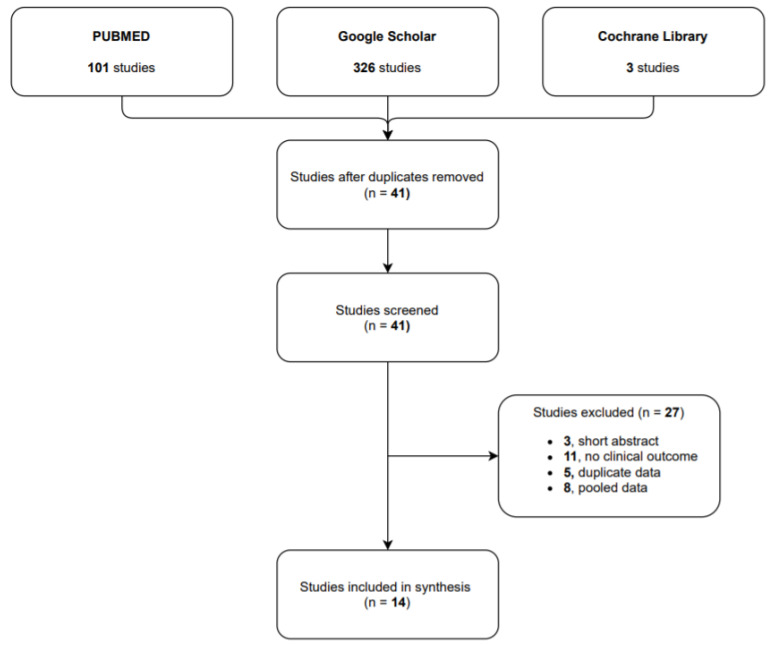
Flow diagram study selection process.

**Table 1 jpm-11-00960-t001:** Study characteristics.

Authors	Country	Study	No. of Patients
Bauchy F et al., 2016 [17]	Belgium	OS	72
Bischoff JM et al., 2014 [18]	Germany	RCT	24
Flöther et al., 2020 [19]	Germany	CR	1
Gaul C et al., 2014 [20]	Germany	CS	2
Giménez-Milà et al., 2014 [21]	Spain	OS	3
Goncalves D et al., 2020 [22]	Portugal	OS	38
Maihofner C et al., 2013 [23]	Germany	OS	238
Mankowski C et al., 2017 [24]	Europe	OS	198
Mullins FC et al., 2021 [25]	Ireland	OS	12
Pinto TJ et al., 2018 [26]	Portugal	OS	28
Roberts C et al., 2012 [27]	USA	CR	1
Tamburini N et al., 2018 [28]	Italy	CS	2
Tiberghien-Chatelain F et al., 2014 [29]	France	OS	12
Zis P et al., 2014 [30]	Greece	CR	1

CR, case report; CS, case series; OS, observational study; RCT, randomized controlled trial.

**Table 2 jpm-11-00960-t002:** Baseline characteristics of patients and CPSP.

Gender, *n* (*%*)	
— Male	109 (38.1)
— Female	177 (61.9)
Age, years	48.4
Localization of pain, *n* (*%*)	
— Head	7 (1.7)
— Chest	85 (20.7)
— Back	7 (1.7)
— Abdomen	52 (12.7)
— Upper limbs	5 (1.2)
— Lower limbs	62 (15.1)
— Other	192 (46.8)

**Table 3 jpm-11-00960-t003:** CPSP aetiology.

Aetiology	No. of Patients	%
Head surgery	2	0.5
Thoracotomy or thoracoscopy	56	15.1
Parietal surgery	42	11.3
Abdominal surgery	4	1.1
Back surgery	6	1.6
Traumatic surgery	5	1.3
Knee surgery	48	12.9
Mastectomy	15	4
Other	194	52.2

**Table 4 jpm-11-00960-t004:** Previous treatment.

Treatment	No. of Studies
Acetaminophen	3
Anticonvulsivants	8
Antidepressants	7
Interventional techniques	1
NSAIDs	2
Opioids	7
Topical lidocaine	3

**Table 5 jpm-11-00960-t005:** Baseline pain intensity.

Authors	Baseline NRS	Baseline DN4
Bauchy F et al., 2016	6 *	4 *
Bischoff JM et al., 2014	4	–
Flöther et al., 2020	7.5	–
Gaul C et al., 2014	9	–
Giménez-Milà et al., 2014	5.7	5.7
Goncalves D et al., 2020	6.2	–
Maihofner C et al., 2013	6.3 *	–
Mankowski C et al., 2017	6.9	–
Mullins FC et al., 2021	7.0	4.5
Pinto TJ et al., 2018	6.3 *	–
Roberts C et al., 2012	8.0	–
Tamburini N et al., 2018	7.0	–
Tiberghien-Chatelain F et al., 2014	7.0	–
Zis P et al., 2014	10	–

* Mixed data.

## Data Availability

Database available in the Department of Woman, Child and General and Specialized Surgery, University of Campania “Luigi Vanvitelli”, Napoli, Italy.

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
