# Peer review of "Capsaicin 8% Patch and Chronic Postsurgical Neuropathic Pain"

_jpm, 2021, doi:10.3390/jpm11100960_

Round 1

Reviewer 1 Report

It’s true, medicine, is in rapid expansion, particularly in the improvement of diagnostic methods as well as the development of new therapies, including those related to personalized medicine.  The authors state that there is a lack of evidence for the treatment of CPSP and in the current study they performed a review of the literature on use of the capsaicin 8% patch to treat neuropathic pain associated with surgery. They aimed to review the efficacy and tolerability of topically applied high-concentration capsaicin for CPSP  in adults.  

Some remarks:

- introduction: Please complete the following aspects: 1. After what type of surgical procedures the incidence of severe CPSP is low (line 36); 2. Give please some details concerning “Various risk factors have been attributed to CPSP”(lines 36-41) and 3. More detailed explanation about the mechanism of action of capsaicin (lines 51-53)

-lines 72-75: the phrase is not clear-line 83: you wrote: “The selected papers were read in full by the two independent reviewers”. These two reviewers are among the authors of this article?-line 117: At first, you can write that you found 101 studies PUBMED, 323 GS…… for a better explanation of Fig 1-lines 164-166: please complete the period of time for which the 14 studies were performed-Table 1: you wrote “No”. It’s better No. of patients. The same for Tables 2 and 3-Table 2: What do you mean by other? 192 from 410 patients fall into the category “Other”. Is very much.At the beginning you wrote that the 14 selected studies reported a total of 632 CPSP cases. At line 170 you wrote: “For 3 studies was not possible to report the patients’ characteristics”. In table 2 for gender you have 286 patients, for localization of pain you have 410 patient. For age how many patients do you have? Therefore, you cannot say that the study is performed on 632 patients.- Table 3: The same as at Table 2: What do you mean by other? 194 from 372 patients fall into the category “Other”. Is very much. Also for CPSP aetiology you don’t have 632 patiens. You have only 372 (including 194 noted as “other”).-Table 4: n=number of studies. The number of patients treated with: Acetaminophen, Anticonvulsivants, Antidepressants etc is not specified. Table 4 should be completed.-line 220: were treated with what? With capsaicin 8% patch? The phrase (lines 220-23) must be correlated with the info presented in Tab 2-Table 5: what represents the star and “mixed data”?-pg 7: Adverse events:    - lines 254-256: You wrote: “It was not possible to determine the number of patients experiencing any type AEs since more than one symptom may appear in an individual participant”. Please reformulate the sentence.    - lines 259-61: an local anesthetic pretreatment is applied. Why? Is more effective than capsaicin 8% patch?    -  line 262: What do you mean by “rescue medications”?- You wrote at lines 17-18; lines 66-67 and lines 338-39: “Our review indicate that capsaicin 8% patch treatment for CPSP is effective, safe and well tolerated”. The informations presented on adverse events are brief (lines 253-266). In my oppinion you can't say that capsaicin 8% patch treatment for CPSP is safe and well tolerated”. -lines 303-305: you wrote: “Of the 632 patients receiving the capsaicin 8% patch in the studies included in this review, 32.3% reported treatment-related AEs”. How could you calculate this percentage considering that only 5 studies have been reported side effects? You reported 623 patients in 14 studies. No adverse effects were present in the other 9 studies? or Side effects were not followed in the 9 studies? Please clarify (lines 253-54 and 303-305).-bibliographic index no. 25 is not complete-line 343: Author contributions: you forgot to report Dr Caterina Aurilio contribution

In conclusion, it is important to demonstrate the effectiveness of applying a patch with capsaicin in order to significantly reduce the CPSP. In this way, the side effects produced by the oral treatments would be avoided. Therefore, the work is welcome.

Taking into account the statement made by the authors at the limitations, namely that: “Recommendations on the effectiveness of capsaicin patch could not be drawn due to the studies design included in our review” I don't know if this review rises to the level of being published in

“Journal of Personalized Medicine” which is ranged as Q1 journal. In my oppinon the Editor  is the one who can decide if the topic of the article is of interest to the Journal readers and if the level of the article is not a bit to low for the claims of the journal.

               My Overall Recommendation for this review is: Accept after minor revision, with the above specification.

Author Response

Reviewer 1

1. After what type of surgical procedures the incidence of severe CPSP is low (line 36) → expecially after thoracotomy, cardiac surgery, hernia surgery, mastectomy and amputations

2. Give please some details concerning “Various risk factors have been attributed to CPSP”(lines 36-41) → We reported a list of risk factors

3. More detailed explanation about the mechanism of action of capsaicin (lines 51-53) → High doses or prolonged exposure to capsaicin induce the defunctionalization of TRPV1. This desensitization involves mechanisms not entirely understood, including depletion of neuropeptides such as substance P in the nerve fibers that express TRPV1 and an increase of intracellular calcium levels.

-lines 72-75: the phrase is not clear → We performed a systematic review based on the Preferred Reporting Items for Systematic Reviews and Meta-Analyses (PRISMA) statement [16]. The protocol was not published but it is available upon request. The review was not registered with the International prospective register of systematic reviews (PROSPERO).

-line 83: you wrote: “The selected papers were read in full by the two independent reviewers”. These two reviewers are among the authors of this article? → The selected papers were read in full by the same two independent reviewers (LGG and PS) and a third reviewer (MCP) was consulted in case of disagreement.

-line 117: At first, you can write that you found 101 studies PUBMED, 323 GS…… for a better explanation of Fig 1 → Our research identified 101 studies in PUBMED, 326 in Google Scholar and 3 in Cochrane Library. After the duplicate studies were removed, we identified 41 articles in the researched databases, and 14 of them were included in our systematic review.

-lines 164-166: please complete the period of time for which the 14 studies were performed → These studies were performed over a period of time from 2012 to 2020.

-Table 1: you wrote “No”. It’s better No. of patients. The same for Tables 2 and 3 → Done

-Table 2: What do you mean by other? 192 from 410 patients fall into the category “Other”. Is very much. → Other or unspecified locations

- At the beginning you wrote that the 14 selected studies reported a total of 632 CPSP cases. At line 170 you wrote: “For 3 studies was not possible to report the patients’ characteristics”. In table 2 for gender you have 286 patients, for localization of pain you have 410 patient. For age how many patients do you have? Therefore, you cannot say that the study is performed on 632 patients. → Due to the nature of the studies, this information was often not reported or could not be obtained with certainty.

- Table 3: The same as at Table 2: What do you mean by other? 194 from 372 patients fall into the category “Other”. Is very much. Also for CPSP aetiology you don’t have 632 patiens. You have only 372 (including 194 noted as “other”). → Other or unspecified aetiologies.

-Table 4: n=number of studies. The number of patients treated with: Acetaminophen, Anticonvulsivants, Antidepressants etc is not specified. Table 4 should be completed. → Due to the nature of the studies, this information was often not reported or could not be obtained with certainty.

-line 220: were treated with what? With capsaicin 8% patch? → The following regions were treated with capsaicin 8% patch:

- The phrase (lines 220-23) must be correlated with the info presented in Tab 2 → As shown in Table 2

-Table 5: what represents the star and “mixed data”? → In these cases authors mixed informations also from other applications.

-pg 7: Adverse events:

- lines 254-256: You wrote: “It was not possible to determine the number of patients experiencing any type AEs since more than one symptom may appear in an individual participant”. Please reformulate the sentence. → The exact number of patients who experienced adverse events could not be determined as more than one AE may appear in a single individual.

- lines 259-61: an local anesthetic pretreatment is applied. Why? Is more effective than capsaicin 8% patch? → Normal procedure to improve the application of the patch

- line 262: What do you mean by “rescue medications”? → medications to give if pain dosen’t improvedespite the application of the patch

- You wrote at lines 17-18; lines 66-67 and lines 338-39: “Our review indicate that capsaicin 8% patch treatment for CPSP is effective, safe and well tolerated”. The informations presented on adverse events are brief (lines 253-266). In my oppinion you can't say that capsaicin 8% patch treatment for CPSP is safe and well tolerated”. → Based on previous studies, capsaicin 8% patch is safe and well tolerated.

-lines 303-305: you wrote: “Of the 632 patients receiving the capsaicin 8% patch in the studies included in this review, 32.3% reported treatment-related AEs”. How could you calculate this percentage considering that only 5 studies have been reported side effects? You reported 623 patients in 14 studies. No adverse effects were present in the other 9 studies? or Side effects were not followed in the 9 studies? Please clarify (lines 253-54 and 303-305). → Where AEs were not reported, we considered them absent.

-bibliographic index no. 25 is not complete → citation from PUBMED “Mullins, C.F., Walsh, S., Rooney, A. et al. A preliminary prospective observational study of the effectiveness of high-concentration capsaicin cutaneous patch in the management of chronic post-surgical neuropathic pain. Ir J Med Sci (2021).

-line 343: Author contributions: you forgot to report Dr Caterina Aurilio contribution → Done

Reviewer 2 Report

The study is aimed to review the literature on the use of the capsaicin 8% patch for the treatment of neuropathic pain associated with surgery.  The title is “Capsaicin 8 % patch and chronic postsurgical neuropathic pain”.

  1. This is a review article.
  2. Several factors influence the treatment. Please discuss these.
  3. Please add more details of the pharmacophysiology of the capsaicin 8% patch.
  4. Please also add the limitations and the adverse effects of the use of capsaicin 8% patch.
  5. What is the new knowledge of the report?
  6. Please recommend to the readers “How to apply this knowledge in clinical practice?”.

Author Response

Reviewer 2

  1. Several factors influence the treatment. Please discuss these. → Several factors influence the treatment: female patients with low baseline pain intensity with the presence of hypoesthesia and no allodynia are more likely to achieve pain relief.

  2. Please add more details of the pharmacophysiology of the capsaicin 8% patch. → High doses or prolonged exposure to capsaicin induce the defunctionalization of TRPV1. This desensitization involves mechanisms not entirely understood, including depletion of neuropeptides such as substance P in the nerve fibers that express TRPV1 and an increase of intracellular calcium levels.

  3. Please also add the limitations and the adverse effects of the use of capsaicin 8% patch. → Treatment with the capsaicin 8 % patch must be performed only by a healthcare provider [11]. Qutenza is not for use near eyes or mucous membranes and it can cause serious side effects, including local pain, redness, itching or increases in blood pressure during or right after treatment [11].

  4. What is the new knowledge of the report? → To date and to our knowledge, however, only very little data is available on the efficacy of high-dose capsaicin therapy in patients with CPSP, although the treatment of neuropathic pain in these patients is a particular challenge due to the complexity and diversity of symptoms. CPSP is challenging to manage because patients are often refractory to or unable to tolerate recommended existing systemic treatments. Our review indicate that 8% capsaicin patch treatment for CPSP is effective, safe and well tolerated, but randomized controlled trials on efficacy, safety and tolerability should be conducted.

  5. Please recommend to the readers “How to apply this knowledge in clinical practice?”. → Our review indicate that 8% capsaicin patch treatment for CPSP is effective, safe and well tolerated, but randomized controlled trials on efficacy, safety and tolerability should be conducted.
